# Assessment of Frailty Scores Among Geriatric Patients Hospitalized in the North-Western Region of Romania: A Cross-Sectional Study

**DOI:** 10.3390/medicina60121947

**Published:** 2024-11-26

**Authors:** Lucreția Avram, Marius I. Ungureanu, Dana Crişan, Valer Donca

**Affiliations:** 1Department of Geriatrics and Gerontology, Iuliu Haţieganu University of Medicine and Pharmacy, 400012 Cluj-Napoca, Romania; avram.lucretia9@gmail.com (L.A.); valerdonca@gmail.com (V.D.); 2Department of Public Health, Babeș-Bolyai University, 400347 Cluj-Napoca, Romania; 3Department of Internal Medicine, Iuliu Haţieganu University of Medicine and Pharmacy, 400012 Cluj-Napoca, Romania; crisan.dc@gmail.com

**Keywords:** Romania, elderly, frailty syndrome, frailty scales, frailty scores

## Abstract

*Background and Objectives:* The global demographic trend of population aging is evident across all regions, with a notable increase in the proportion of elderly individuals. Romania exemplifies this phenomenon, as 17% of its population is currently aged 65 years or older—a figure projected to rise to 25% by 2050. This demographic shift underscores the pressing need for comprehensive measures to address the health and social requirements of this growing population segment. This study aims to assess the prevalence of frailty among older adults in Romania and explore its relationship with socioeconomic factors. *Materials and Methods:* We employed a quantitative approach, by using cross-sectional data from patients hospitalized at the geriatrics ward of the Municipal Clinical Hospital in Cluj-Napoca, Romania. Frailty scores were calculated through established frailty assessment tools, allowing for a comprehensive evaluation of frailty status. In addition, we compared the socioeconomic characteristics of frail and non-frail patients to identify potential disparities. Statistical analyses were performed to assess associations between frailty and socioeconomic factors, providing insight into the relationship between these variables within the patient population. *Results:* The prevalence of frailty in our sample is, depending on the frailty scale used, 55% to 79%, which is in line with figures from specialized geriatric wards in other studies. There is moderate to substantial agreement between the scales we compared, and all six scales seem to concurrently agree on the frailty diagnostic in 55% of cases. Additionally, frail patients are more likely to have a low socioeconomic status. *Conclusions:* A significant limitation in European frailty research has been the absence of comparative frailty prevalence data across several European countries, especially those with lower economic development. Our study fills this gap by providing data on frailty prevalence in the north-western region of Romania.

## 1. Introduction

The global trend of population aging has become increasingly evident, particularly in high-income countries, where the proportion of elderly individuals has grown significantly. This phenomenon is now visible in other parts of the world, including Romania, where 17% of the population is aged 65 or older. Projections indicate that by 2050, this figure will rise to 25% [1]. These demographic changes underscore the urgent need to develop appropriate health and social care strategies to address the complex needs of the aging population. Without such measures, older individuals face a heightened risk of developing illnesses, disabilities, and dependencies, which could lead to increased healthcare costs and significant socioeconomic pressures [2].

Frailty, an age-associated syndrome, represents a critical challenge in this context. It is characterized by a substantial reduction in physiological reserves and resilience, making affected individuals highly vulnerable to adverse health outcomes even in response to minor stressors. Unlike the gradual physical decline typically associated with aging, frailty represents an accelerated loss of functionality and stability [3,4]. Global studies indicate that frailty affects between 12% and 24% of older adults, with higher prevalence rates observed in low- and middle-income countries compared to high-income countries. The unequal distribution of frailty research highlights a critical gap, especially in the context of health systems that are less prepared to manage the complexity of this condition [5,6]. The multidimensional nature of frailty necessitates comprehensive assessment tools that integrate physical, psychological, and social components. While widely used tools, such as the Fried Frailty Phenotype and the Rockwood Frailty Index, have provided valuable insights, their complexity and the resources required for implementation hinder their routine application in clinical practice [7,8,9]. Additionally, existing tools often focus on predicting health outcomes, such as morbidity or mortality, rather than directly addressing frailty itself. The concept of “healthy aging”, promoted by the World Health Organization (WHO), emphasizes the need to shift from a deficit-based approach to one that prioritizes maintaining functional ability throughout life [5,6].

Studies on frailty have revealed significant variations in prevalence estimates, influenced by the methodology used and the population studied. Large-scale initiatives such as the Survey of Health and Retirement in Europe (SHARE) project have provided valuable insights into frailty across European countries [10]. However, these studies often exclude institutionalized individuals and rely on complex data collection methods, limiting their applicability in local contexts. Moreover, the tools used frequently emphasize predicting health outcomes, such as morbidity or mortality, rather than addressing frailty directly [11,12].

In Romania, the challenges of managing frailty are compounded by limited research, a lack of standardized assessment tools, and insufficient awareness among healthcare professionals. These gaps hinder the early identification of frailty and the implementation of targeted interventions, thereby exacerbating the burden on the healthcare system [13,14].

This study aims to address these needs by investigating the prevalence of frailty in a sample of elderly patients from Romania using multiple assessment tools to evaluate their effectiveness and agreement. Additionally, the study explores the relationship between frailty and socioeconomic factors, providing insights into the local determinants of frailty. To contribute to these efforts, we propose the GerEVal Napoca Frailty Index (GVN-FI), a new tool designed to meet the specific needs within the Romanian healthcare context and facilitate the better integration of frailty assessment into clinical practice.

## 2. Materials and Methods

This cross-sectional study included 516 patients aged 55 years and older who were hospitalized in the Geriatrics Ward of the Municipal Clinical Hospital in Cluj-Napoca, Romania, between 1 January 2023 and 31 January 2024. Initially, 610 patients were recruited, but 94 were excluded based on the application of exclusion criteria. These criteria included patients with acute conditions, severe dementia, sepsis, or those requiring intensive care, as well as cases where not all parameters of the frailty scales could be assessed.

Admissions were made through referrals from general practitioners for chronic conditions and were scheduled in advance. This study also included patients aged 55–64 years, acknowledging that the aging process begins well before this age. These patients often presented with complex comorbidities, early-onset frailty, or accelerated aging caused by chronic diseases and functional deficits. This interdisciplinary approach, specific to geriatrics, was justified by the need to prevent complications and manage complex health issues [15,16]. During hospitalization, a comprehensive geriatric assessment (CGA) was performed for each patient, capturing detailed socioeconomic (e.g., age, sex, net pension), biochemical (e.g., complete blood count, albumin, C-reactive protein), and anthropometric data (e.g., weight, height). Furthermore, physical and psychological functioning were assessed using standardized tools, including frailty scales, walking speed, the SARC-F test for sarcopenia, and the Montreal Cognitive Assessment (MoCA) test. For this study, the analysis focuses on six frailty assessment tools: the GVN-FI, a frailty index developed from CGA data; the frailty index developed by Beth Israel Deaconess Medical Center (BIDMC-FI) [17], the Clinical Frailty Scale (CFS) [18], the Edmonton Frailty Scale (EFS), the Fried Frailty Phenotype (FP), and the Vivifrail Test (VT) [19]. An overview of the frailty scales is available in Table 1.

For EFS, we considered the category “Vulnerable” (scores 4–5) to be equal to pre-frailty. For some of the scales, there are multiple cut-off points for frailty in use. In such cases, we defaulted to using the more popular cut-off points. For instance, we chose to define the frailty threshold for the EFS based on scores above 5, although some authors define it based on scores above 7.

For the GVN-FI, a total of 46 variables were assessed in hospitalized patients (see Table A1 from Appendix A). These self-reported items encompassed a comprehensive range of health-related indicators, including symptoms, health attitudes, medical conditions, and impaired functioning. Most of the variables were categorical and binary, and we assigned values on a scale from 0 to 1, where a higher value indicated an increased risk of frailty. For the study’s applicability, a minimum of 30 items was required, and the frailty index was calculated as the ratio of the number of present deficits to the total number of deficits considered/applied to the patient.

We have developed this new frailty index following an extensive literature review, drawing upon the work of Searle and colleagues [20]. Their cohort study, conducted in New Haven, Connecticut, involved the enrollment of individuals aged 70 years and older. Specifically, 754 community-dwelling, non-disabled individuals with a life expectancy exceeding 12 months were recruited. Comprehensive assessments were conducted at participants’ homes at baseline and subsequently every 18 months. At the 18-month follow-up, 681 participants, aged 72 to 98, were evaluated. This report utilizes both the baseline and 18-month follow-up data to facilitate comparisons between them and to examine how the newly created frailty indexes in this dataset align with previously established ones. Mortality was monitored monthly for a duration of nine years from the initial interview and confirmed through obituaries and death certificates. This approach to quantifying frailty offers valuable insights into health and frailty-related characteristics and outcomes in older adults.

### Statistical Analysis

Descriptive statistics for the patients, stratified by sex and by frailty status, have been computed, along with the prevalence of frailty by sex and by type of scale. We used Pearson’s Chi-squared tests, Fisher’s exact tests, or Wilcoxon rank sum tests to explore if there are statistically significant differences between the groups. We chose to use BIDMC-FI as the main indicator of frailty because it is the most comprehensive scale among the ones used in this study. The level of agreement between pairs of scales has been computed using Cohen’s kappa coefficient. In addition, we used Fleiss’ kappa coefficient to assess the level of agreement across all of the scales. For agreement, we used binary results (frail and not frail) instead of ternary results (frail, pre-frail, robust), defining all scores under the frailty threshold as “not frail”. In cases where the scale made distinctions between frailty severity (e.g., mild, moderate, severe frailty) we aggregated all categories under a single “frailty” category. Missing data was minimal, comprising less than 6% per variable. All statistical analyses were performed using R program 4.2.2 [21].

## 3. Results

In terms of frailty diagnosis, there was moderate (0.40–0.60) or substantial (0.61–0.80) agreement between the scales we compared. Likewise, all six scales seemed to concurrently agree on the frailty diagnostic in 55% of cases (Fleiss’ kappa: 0.546). More information is available in Table 2. The strongest agreement of the new index was observed with the BIDMC-FI, CFS, and EFS, whereas with the FP and VT scales, a value below 0.50 was obtained; this lower value may potentially be attributed to the group size and the distribution of evaluations. This validation demonstrates that the tool is effective, easy to obtain from a careful medical history and physical exam, and easy to apply.

The prevalence of frailty in this sample ranged from 55% to 79%, depending on the scale used. While the greater prevalence of frailty among women is largely recognized by literature, in this sample, significant differences by sex were observed only for the EFS and the VT (see Table 3).

In terms of socioeconomic status, a large proportion of the patients (38%) came from rural areas, and roughly half of the whole sample (54%) had not finished high school. There was a large variability in terms of age and pension, with values between 55 and 99 years and between 0 and 16,000 lei, respectively. It is also worth mentioning that only 20 patients in our sample were under 65 years old.

In this sample, there were statistically significant differences in terms of education and pension between men and women: women were, on average, less educated (*p* < 0.001) and received a smaller pension (μ_women_ = 2020.4, μ_men_ = 3074.1, *p* < 0.001). Frail patients (according to BIDMC-FI) were, on average, less educated than non-frail patients (*p* < 0.001); only around 41% of them finished high school, in contrast to around 60% of non-frail patients. Frail patients also tended to be older (μ_frail_ = 81.2, μ_non-frail_ = 74.6, *p* < 0.001) and to have a smaller net pension (μ_frail_ = 2274.4, μ_non-frail_ = 2523.2, *p* < 0.05). There were also statistically significant differences by sex and by frailty status for marital status; women and frail patients were more likely to be widows (*p* < 0.001). However, these differences might be accounted for by the fact that women tend to live longer than men, and frail people tend to be older than non-frail people. Complete results are available in Table 4 and Table 5.

## 4. Discussion

Our study reveals several important findings about the elderly population in north-west Romania, with a focus on socioeconomic factors, education, and pension disparities. The sample showed a high degree of variability, particularly in age, educational attainment, and pension income. Notably, only 38% of the patients came from rural areas, and a significant portion (54%) had not completed high school. There were also distinct differences between men and women, as well as between frail and non-frail patients. Women were, on average, less educated and received smaller pensions than men. Similarly, frail patients tended to be older, less educated, and had lower pensions than their non-frail counterparts. These findings highlight the complex interplay between socioeconomic status, gender, and frailty in this population, which, to date, has been insufficiently documented in the scientific literature for the elderly in Romania [22,23]. Moreover, the fact that only 38% of the patients were dwelling in rural areas might point to the limited access of rural patients to hospital services, since roughly half of the Romanian population resides in rural settings. This finding suggests that rural elderly individuals might be particularly vulnerable to health decline due to the lack of available resources and services that are likely more accessible for those living in urban areas.

While the prevalence of frailty in our sample (55–79%) is much higher than the prevalence of frailty in the general population across the world [16], it is in line with figures from specialized geriatric wards in other studies, where the pooled prevalence is 66.5% (95% CI: 54.3–78.7%) [24]. However, our study also highlights some variability in the frailty assessment tools used. Although moderate to substantial agreement was observed between the different scales (Fleiss’ kappa: 0.546), the wide range of frailty prevalence (55% to 79%) across these tools suggests that they may measure frailty differently. This inconsistency can complicate clinical decision-making, as it is unclear which tool offers the most accurate diagnosis for this population. Moreover, these scales may not fully account for the specific cultural and contextual factors relevant to the Romanian elderly population, which could affect their applicability. As such, our decision was to develop the GerEVal Napoca Frailty Index. Definitions associated with frailty are continually being adjusted, similarly to other diseases, emphasizing the need to develop an efficient and practical tool for clinical use. The proposed new approach, which is based on the assessment of deficit accumulation, makes a significant contribution to the understanding and management of frailty. It recognizes that, on average, people accumulate deficits as they age, but a challenging aspect is that these individuals do not accumulate the same types of deficits, nor do they do so at the same rate. In this light, the newly proposed frailty index has the advantage of counting deficits without excessively focusing on their nature, thus allowing for a more comprehensive evaluation of the health status of elderly patients. In the conducted research, it was found that the number of deficits is a significant indicator of health risks: the more deficits a person accumulates, the more exposed they are to the risks associated with frailty. This index not only highlights the correlation between deficit accumulation and the loss of physiological reserves but also facilitates the early identification of vulnerable patients, enabling faster and more effective interventions. The adoption of this tool in clinical practice not only improves the health management of the elderly but also contributes to the optimization of resources within the Romanian healthcare system. One of the main benefits of this index is how quickly it can be applied. Much of the necessary information for the assessment can be obtained during a careful medical history and physical exam. This makes it easy to include in daily clinical practice, allowing doctors to quickly assess the health status of their elderly patients. This will be particularly useful in the Romanian context, where the count of hospital beds has long been the primary unit of assessing healthcare performance. With the new tool, we also suggest that a new paradigm be adopted by the health and care system, in which the assessment of frailty within geriatric evaluation becomes essential for determining patient needs. This evaluation aims to identify individual resources and capacities and to develop a person-centered long-term treatment and follow-up plan respectful of the individual’s preferences, needs, and values.

The study’s findings carry significant implications for healthcare policy and clinical practice, emphasizing the need to address frailty, socioeconomic inequalities, and gender disparities among elderly populations, particularly non-hospitalized elderly individuals. For improving clinical practice, clinicians should prioritize early detection and management of frailty through comprehensive geriatric assessments and personalized care plans. This approach should integrate both physical and mental health interventions, recognizing the interconnected nature of these factors in aging populations. Moreover, from a policy perspective, addressing the unique needs of vulnerable groups will require targeted and multifaceted healthcare programs. Public health interventions should prioritize improving access to healthcare, educational opportunities about healthy aging, and financial security for elderly individuals, especially those in rural areas with limited resources. Gender-specific strategies are also crucial, as female elderly patients often face higher levels of socioeconomic vulnerability, compounded by disparities in healthcare access and outcomes.

Investing in community-based health services, such as mobile clinics and telemedicine, can bridge the gap for rural populations, while programs aimed at empowering elderly women through financial literacy and social support can foster resilience. By tailoring these interventions to the specific socioeconomic, geographic, and gender-related needs of elderly individuals, policymakers can effectively reduce frailty, enhance independence, and improve the overall quality of life for Romania’s aging population. This comprehensive approach not only aligns with the goals of equitable healthcare but also strengthens the sustainability of health systems in the context of an aging society.

### Limitations

Whereas our study offers valuable insights into the frailty and socioeconomic status of elderly patients in north-western Romania, it also presents several limitations.

First, while the sample size was sufficient for statistical analysis, it may not be fully representative of the broader elderly population, particularly those who are not hospitalized. The study is limited to patients who were admitted to the Cluj-Napoca Municipal Hospital, which could lead to an overrepresentation of individuals with more severe health conditions. This selection bias might skew the findings and limit the generalizability of the results to healthier elderly individuals, especially those living independently in the community. Additionally, geographic factors may have influenced patient inclusion, as those from rural areas with limited healthcare access might be underrepresented in the study.

Another key limitation is the cross-sectional design, which makes it difficult to establish causal relationships between socioeconomic status and frailty. While the study identified significant associations between education, pension, and frailty, the direction of these relationships remains unclear. For instance, it is uncertain whether low socioeconomic status leads to frailty or if frailty contributes to worsened economic conditions. A longitudinal approach would provide more robust insights into how these factors interact over time, offering a clearer understanding of the causal mechanisms at play. Sociocultural factors, such as family support and community engagement, were not directly assessed in the study, although these may play a significant role in the health and well-being of elderly individuals, particularly in rural areas. These unmeasured factors could moderate the relationship between socioeconomic status and frailty, and their absence limits the depth of the analysis.

Furthermore, the study’s reliance on pension data as a measure of economic status introduces the possibility of reporting errors. While pension data offer an important economic indicator, they do not capture a more comprehensive view of financial resources, such as wealth, assets, or changes in income over time. A more detailed economic analysis could provide greater insight into how financial stability influences frailty.

The study focuses solely on patients admitted to the Cluj-Napoca Municipal Hospital, potentially overrepresenting individuals with more severe health conditions. This selection bias could distort the findings and limit their applicability to healthier elderly populations, particularly non-hospitalized elderly individuals living independently in the community. Furthermore, geographic factors may have influenced patient inclusion, possibly leading to an underrepresentation of those from rural areas with limited access to healthcare services.

Despite these limitations, our study does contribute to the understanding of frailty in north-western Romania, where comprehensive data is lacking, and it highlights areas for future research, particularly in addressing the socioeconomic determinants of frailty and refining diagnostic tools for elderly patients.

## 5. Conclusions

Frailty serves as a comprehensive indicator of health status among the elderly. A significant limitation in European frailty research has been the absence of comparative frailty prevalence data across several European countries, especially those with lower economic development. Our study fills this gap by providing data on frailty prevalence in the north-western region of Romania. We hope our findings will encourage researchers to consider frailty as a unified measurable metric to assess disparities in population characteristics across countries. This approach could support the refinement of European policies regarding healthcare system development, research funding, and structural grants, ultimately contributing to reducing the persistent health disparities between countries with varying levels of economic development.

## Figures and Tables

**Table 1 medicina-60-01947-t001:** Overview of frailty scales.

Scale	Score Range	Categories
Fried Frailty Phenotype (FP)	0–5	0 = Robust
		1–2 = Pre-fail
		3–5 = Frail
Clinical Frailty Scale (CFS)	1–9	1–3 = Robust
		4 = Pre-frail
		5–8 = Frail
		9 = Terminally ill
GerEVal Napoca Frailty Index (GVN-FI)	0.000–0.450+	0.000–0.149 = Robust
		0.150–0.299 = Pre-fail
		0.300–0.449 = Frail
		0.450+ = Very frail
Beth Israel Deaconess Medical Center Frailty Index (BIDMC-FI)	0.000–0.550+	0.000–0.149 = Robust
		0.150–0.249 = Pre-frailty
		0.250–0.349 = Mild frailty
		0.350–0.449 = Moderate frailty
		0.450–0.549 = Severe frailty
		0.550+ = Advanced frailty
The Edmonton Frailty Scale (EFS)	0–17	0–3 = Fit
		4–5 = Vulnerable
		6–7 = Mild frailty
		8–9 = Moderate frailty
		10+ = Severe frailty
Vivifrail Test (VT)	0–12	0–3 = Type A (Disability)
		4–6 = Type B (Frailty)
		7–9 = Type C (Pre-frailty)
		10–12 = Type D (Robust)

**Table 2 medicina-60-01947-t002:** Cohen’s kappa coefficient of agreement for frailty scales.

	BIDMC-FI	GVN-FI	CFS	EFS	FP	VT	
**BIDMC-FI**		0.584	0.646	0.585	0.536	0.610	
**GVN-FI**	0.584		0.668	0.636	0.434	0.483	
**CFS**	0.646	0.668		0.652	0.498	0.526	**Fleiss’ kappa:** **0.546**
**EFS**	0.585	0.636	0.652		0.441	0.470	
**FP**	0.536	0.434	0.498	0.441		0.636	
**VT**	0.610	0.483	0.526	0.470	0.636		

**Table 3 medicina-60-01947-t003:** Distribution of results across frailty scales.

	Overall (N = 516)	Women (N = 358)	Men (N = 158)	
Characteristic	N (%)	N (%)	N (%)	*p* ^1,2^
Beth Israel Deaconess Medical Center Frailty Index				ns
Robust	74 (14.9)	51 (14.8)	23 (15.3)	
Pre-frail	87 (17.6)	56 (16.2)	31 (20.7)	
Frail	334 (67.5)	238 (69.0)	96 (64.0)	
GerEVal Napoca Frailty Index				ns
Robust	15 (2.9)	7 (2.0)	8 (5.1)	
Pre-frail	94 (18.4)	61 (17.2)	33 (20.9)	
Frail	403 (78.7)	286 (80.8)	117 (74.0)	
Clinical Frailty Scale				ns
Robust	58 (11.3)	35 (9.8)	23 (14.6)	
Pre-frail	72 (14.0)	45 (12.6)	27 (17.1)	
Frail	385 (74.7)	277 (77.6)	108 (68.3)	
Edmonton Frailty Scale				***
Robust	38 (7.8)	26 (7.6)	12 (8.2)	
Pre-frail	81 (16.6)	42 (12.3)	39 (26.7)	
Frail	368 (75.6)	273 (80.1)	95 (65.1)	
Fried Frailty Phenotype				ns
Robust	46 (8.9)	31 (8.7)	15 (9.5)	
Pre-frail	187 (36.4)	125 (35.1)	62 (39.2)	
Frail	281 (54.7)	200 (56.2)	81 (51.3)	
Vivifrail Test				*
Robust	87 (16.9)	53 (14.8)	34 (21.7)	
Pre-frail	132 (25.6)	86 (24.0)	46 (29.3)	
Frail	296 (57.5)	219 (61.2)	77 (49.0)	

Note: Missing observations excluded; ^1^ ns = not significant at the 0.05 level; * *p* < 0.05; *** *p* < 0.001 ^2^ Pearson’s Chi-squared test; Fisher’s exact test; Wilcoxon rank sum test.

**Table 4 medicina-60-01947-t004:** Descriptive statistics by sex.

	Total (N = 516)	Women (N = 358)	Men (N = 158)	
Characteristic	N (%)	N (%)	N (%)	*p* ^1,2^
Living location				ns
Urban	318 (61.6)	218 (60.9)	100 (63.3)	
Rural	198 (38.4)	140 (39.1)	58 (36.7)	
Education				***
No school	4 (0.8)	3 (0.8)	1 (0.6)	
Elementary school	96 (18.6)	80 (22.3)	16 (10.1)	
Middle school	177 (34.2)	127 (35.6)	50 (31.6)	
High school	166 (32.2)	109 (30.4)	57 (36.1)	
University	70 (13.6)	38 (10.6)	32 (20.3)	
Postgraduate	3 (0.6)	1 (0.3)	2 (1.3)	
Marital status				***
Married	187 (36.5)	94 (26.5)	93 (59.2)	
Widow	293 (57.3)	240 (67.6)	53 (33.8)	
Divorced	21 (4.1)	15 (4.2)	6 (3.8)	
Unmarried	11 (2.1)	6 (1.7)	5 (3.2)	
	Mean (sd); Median (min-max)	Mean (sd); Median (min-max)	Mean (sd); Median (min-max)	
Age	79.1 (7.8); 79.5 (55.4–99.9)	79.1 (7.6); 79.5 (55.4–93.3)	79.1 (8.2); 79.4 (59.5–99.9)	ns
Pension (lei)	2342.7 (1305.9); 2136.0 (0–16,000)	2020.4 (864.3); 1853.5 (113–5400)	3074.1 (1767.1); 2837.0 (0–16,000)	***

Note: Missing observations excluded; ^1^ ns = not significant at the 0.05 level; *** *p* < 0.001; ^2^ Pearson’s Chi-squared test; Fisher’s exact test; Wilcoxon rank sum test.

**Table 5 medicina-60-01947-t005:** Descriptive statistics by frailty status (BIDMC-FI).

	Not Frail (N = 161)	Frail (N = 334)	
Characteristic	N (%)	N (%)	*p* ^1,2^
Sex			ns
Women	107 (66.5)	238 (71.3)	
Men	54 (33.5)	96 (28.7)	
Living location			ns
Urban	103 (64.0)	207 (62.0)	
Rural	58 (36.0)	127 (38.0)	
Education			***
No school	1 (0.6)	3 (0.9)	
Elementary school	16 (9.9)	74 (22.2)	
Middle school	48 (29.8)	121 (36.2)	
High school	63 (39.2)	98 (29.3)	
University	31 (19.3)	37 (11.1)	
Postgraduate	2 (1.2)	1 (0.3)	
Marital status			***
Married	70 (43.5)	109 (33.0)	
Widow	74 (45.9)	206 (62.5)	
Divorced	14 (8.7)	7 (2.1)	
Unmarried	3 (1.9)	8 (2.4)	
	Mean (sd); Median (min–max)	Mean (sd); Median (min–max)	
Age	74.6 (8.0); 73.6 (55.4–93.3)	81.2 (6.7); 81.6 (63.3–99.9)	***
Pension (lei)	2523.2 (1324.2); 2247.0 (149–7872)	2274.4 (1317.4); 2020.5 (0–16,000)	*

Note: Missing observations excluded; ^1^ ns = not significant at the 0.05 level; * *p* < 0.05; *** *p* < 0.001; ^2^ Pearson’s Chi-squared test; Fisher’s exact test; Wilcoxon rank sum test.

## Data Availability

The dataset used during the current study is available from the corresponding authors upon reasonable request.

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
