# Peer review of "Assessment of Frailty Scores Among Geriatric Patients Hospitalized in the North-Western Region of Romania: A Cross-Sectional Study"

_medicina, 2024, doi:10.3390/medicina60121947_

Round 1
Reviewer 1 Report
Comments and Suggestions for Authors
The study addresses an important and timely issue of frailty among the elderly, particularly in a context that has been under-researched. The manuscript is well-structured, with clear sections for the abstract, introduction, methods, results, discussion, and conclusions. However, some sections could benefit from more concise language to enhance readability.
l it would be beneficial to include more recent references, particularly studies published in the last two years, to ensure the literature review is current.
l The abstract effectively summarizes the study. However, it could be more concise, particularly in the methods section. Consider focusing on the key aspects of the methodology rather than detailing all scales used.
l The methods section is comprehensive. However, it would be helpful to clarify how participants were selected and any inclusion/exclusion criteria applied. Consider providing more detail on the statistical analysis methods used, including any software specifics (e.g., version of R).
l The discussion effectively contextualizes the findings within existing literature. However, it could delve deeper into the implications of the findings for clinical practice and policy. Consider discussing limitations more thoroughly, particularly regarding the generalizability of the findings to non-hospitalized elderly populations.
Thank you for the opportunity to review this important work.
Author Response
Comments 1: l it would be beneficial to include more recent references, particularly studies published in the last two years, to ensure the literature review is current
Response 1: We have updated the bibliography
Comments 2: The abstract effectively summarizes the study. However, it could be more concise, particularly in the methods section. Consider focusing on the key aspects of the methodology rather than detailing all scales used.
Response 2: Background: The global demographic trend of population aging is evident across all regions, with a notable increase in the proportion of elderly individuals. Romania exemplifies this phenomenon, as 17% of its population is currently aged 65 years or older—a figure projected to rise to 25% by 2050. This demographic shift underscores the pressing need for comprehensive measures to address the health and social requirements of this growing population segment. This study aims to assess the prevalence of frailty among older adults in Romania and explore its relationship with socioeconomic factors. Methods: We employed a quantitative approach, by using cross-sectional data from patients hospitalised at the Geriatrics ward of the Municipal Clinical Hospital in Cluj-Napoca, Romania. Frailty scores were calculated through established frailty assessment tools, allowing for a comprehensive evaluation of frailty status. In addition, we compared the socioeconomic characteristics of frail and non-frail patients to identify potential disparities. Statistical analyses were performed to assess associations between frailty and socioeconomic factors, providing insight into the relationship between these variables within the patient population.
Comments 3: The methods section is comprehensive. However, it would be helpful to clarify how participants were selected and any inclusion/exclusion criteria applied. Consider providing more detail on the statistical analysis methods used, including any software specifics (e.g., version of R).
Response 3: This cross-sectional study included 516 patients aged 55 years and older who were hospitalized in the Geriatrics Ward of the Municipal Clinical Hospital in Cluj-Napoca, Romania, between January 1, 2023, and January 31, 2024. Initially, 610 patients were recruited, but 94 were excluded based on the application of exclusion criteria. These criteria included patients with acute conditions, severe dementia, sepsis, or those requiring intensive care, as well as cases where not all parameters of the frailty scales could be assessed. Admissions were made through referrals from general practitioners for chronic conditions and were scheduled in advance.
All statistical analyses were performed using R program 4.2.2.
Comments 4: The discussion effectively contextualizes the findings within existing literature. However, it could delve deeper into the implications of the findings for clinical practice and policy. Consider discussing limitations more thoroughly, particularly regarding the generalizability of the findings to non-hospitalized elderly populations.
Response 4: The study’s findings carry significant implications for healthcare policy and clinical practice, emphasizing the need to address frailty, socioeconomic inequalities, and gender disparities among elderly populations, particularly non-hospitalized elderly individuals. For improving clinical practice, clinicians should prioritize early detection and management of frailty through comprehensive geriatric assessments and personalized care plans. This approach should integrate both physical and mental health interventions, recognizing the interconnected nature of these factors in ageing populations. Moreover, from a policy perspective, addressing the unique needs of vulnerable groups will require targeted and multifaceted healthcare programs. Public health interventions should prioritize improving access to healthcare, educational opportunities about healthy ageing, and financial security for elderly individuals, especially those in rural areas with limited resources. Gender-specific strategies are also crucial, as female elderly patients often face higher levels of socioeconomic vulnerability, compounded by disparities in healthcare access and outcomes. Investing in community-based health services, such as mobile clinics and telemedicine, can bridge the gap for rural populations, while programs aimed at empowering elderly women through financial literacy and social support can foster resilience. By tailoring these interventions to the specific socioeconomic, geographic, and gender-related needs of elderly individuals, policymakers can effectively reduce frailty, enhance independence, and improve the overall quality of life for Romania’s ageing population. This comprehensive approach not only aligns with the goals of equitable healthcare but also strengthens the sustainability of health systems in the context of an ageing society.
The study focuses solely on patients admitted to the Cluj-Napoca Municipal Hospital, potentially overrepresenting individuals with more severe health conditions. This selection bias could distort the findings and limit their applicability to healthier elderly populations, particularly non-hospitalized elderly individuals living independently in the community. Furthermore, geographic factors may have influenced patient inclusion, possibly leading to an underrepresentation of those from rural areas with limited access to healthcare services.
Reviewer 2 Report
Comments and Suggestions for Authors
Thank you for the opportunity to review the study entitled “Assessment of frailty scores among hospitalised elderly from the North-Western region in Romania: a cross-sectional study.” As the authors indicate, their study fills this gap by providing data on frailty prevalence in the North-Western region of Romania. Addressing this topic is of great value and provides insight into the prevalence of frailty within the Romanian population. However, the topic of the paper should be modified, as the authors mention in the methodology section that the data were collected from patients hospitalised in the geriatric department. Therefore, the studied population more accurately reflects the geriatric population rather than elderly patients hospitalised in general. The title of the paper should reflect this distinction. On the other hand, the researchers mention in the methodology section that 516 patients aged 55 or older were included in the study. Later, however, they state in the results section that only 20 patients out of 230 in the sample were under 55 years old. These data appear inconsistent and require clarification. Additionally, it is unclear why patients aged 55 were hospitalised in geriatric wards, as this does not align with the typical definition of old age.
The introduction of the manuscript is too lengthy, and this is not the place to describe the specific tools used to assess frailty, as the authors have done for the Frailty Phenotype. It lacks an emphasis that frailty is not synonymous with old age and that it is a multidimensional construct. The authors should highlight that studies may underestimate the prevalence of frailty in certain populations because most frailty assessment tools are unidimensional. The authors of the manuscript also primarily use tools that assess the physical dimension of frailty. Developing their own tool is an asset of the paper; however, it is composed of existing tools already available for clinical use. Furthermore, it is lengthy and relies on both subjective data and several objective measures. Nonetheless, the tool is more time-consuming to implement in clinical practice than other available frailty assessment tools in the literature.
Author Response
Comments 1: The topic of the paper should be modified, as the authors mention in the methodology section that the data were collected from patients hospitalised in the geriatric department. Therefore, the studied population more accurately reflects the geriatric population rather than elderly patients hospitalised in general. The title of the paper should reflect this distinction.
Response 1: Thank you. We empathize with you, which is why we have chosen the title: "Assessment of Frailty Scores Among Geriatric Patients Hospitalised in the North-Western Region of Romania: A Cross-Sectional Study''
Comments 2: The researchers mention in the methodology section that 516 patients aged 55 or older were included in the study. Later, however, they state in the results section that only 20 patients out of 230 in the sample were under 55 years old. These data appear inconsistent and require clarification. Additionally, it is unclear why patients aged 55 were hospitalised in geriatric wards, as this does not align with the typical definition of old age.
Response 2: This cross-sectional study included 516 patients aged 55 years and older who were hospitalized in the Geriatrics Ward of the Municipal Clinical Hospital in Cluj-Napoca, Romania, between January 1, 2023, and January 31, 2024. Initially, 610 patients were recruited, but 94 were excluded based on the application of exclusion criteria. These criteria included patients with acute conditions, severe dementia, sepsis, or those requiring intensive care, as well as cases where not all parameters of the frailty scales could be assessed. Admissions were made through referrals from general practitioners for chronic conditions and were scheduled in advance. Although geriatrics typically addresses individuals aged 65 years and older, this study also included patients aged 55–64 years, acknowledging that the aging process begins well before this age. Notably, at a general population level, a significant proportion of patients under 65 years are identified as pre-frail, underscoring the need for targeted interventions to address this early stage of frailty. These patients often presented with complex comorbidities, early-onset frailty, or accelerated aging caused by chronic diseases and functional deficits. This interdisciplinary approach, specific to geriatrics, was justified by the need to prevent complications and manage complex health issues. Furthermore, resource constraints and limited access to other specialties, such as internal medicine or rehabilitation, led to the allocation of these patients to geriatrics, adapting to local healthcare realities.
It is also worth mentioning that only 20 patients in our sample were under 65 years old ( our mistake).
Comments 3: The introduction of the manuscript is too lengthy, and this is not the place to describe the specific tools used to assess frailty
Response 3: The global trend of population aging has become increasingly evident, particularly in high-income countries, where the proportion of elderly individuals has grown significantly. This phenomenon is now visible in other parts of the world, including Romania, where 17% of the population is aged 65 or older. Projections indicate that by 2050, this figure will rise to 25%. These demographic changes underscore the urgent need to develop appropriate health and social care strategies to address the complex needs of the aging population. Without such measures, older individuals face a heightened risk of developing illnesses, disabilities, and dependencies, which could lead to increased healthcare costs and significant socioeconomic pressures. Frailty, an age-associated syndrome, represents a critical challenge in this context. It is characterized by a substantial reduction in physiological reserves and resilience, making affected individuals highly vulnerable to adverse health outcomes even in response to minor stressors. Unlike the gradual physical decline typically associated with aging, frailty represents an accelerated loss of functionality and stability. Global studies indicate that frailty affects between 12% and 24% of older adults, with higher prevalence rates observed in low- and middle-income countries compared to high-income countries. The unequal distribution of frailty research highlights a critical gap, especially in the context of health systems that are less prepared to manage the complexity of this condition. The multidimensional nature of frailty necessitates comprehensive assessment tools that integrate physical, psychological, and social components. While widely used tools, such as the Fried Frailty Phenotype and the Rockwood Frailty Index, have provided valuable insights, their complexity and the resources required for implementation hinder their routine application in clinical practice. Additionally, existing tools often focus on predicting health outcomes, such as morbidity or mortality, rather than directly addressing frailty itself. The concept of "healthy aging," promoted by the World Health Organization (WHO), emphasizes the need to shift from a deficit-based approach to one that prioritizes maintaining functional ability throughout life. Studies on frailty have revealed significant variations in prevalence estimates, influenced by the methodology used and the population studied. Large-scale initiatives such as the SHARE (Survey of Health and Retirement in Europe) project have provided valuable insights into frailty across European countries. However, these studies often exclude institutionalized individuals and rely on complex data collection methods, limiting their applicability in local contexts. Moreover, the tools used frequently emphasize predicting health outcomes, such as morbidity or mortality, rather than addressing frailty directly. In Romania, the challenges of managing frailty are compounded by limited research, a lack of standardized assessment tools, and insufficient awareness among healthcare professionals. These gaps hinder the early identification of frailty and the implementation of targeted interventions, thereby exacerbating the burden on the healthcare system. This study aims to address these needs by investigating the prevalence of frailty in a sample of elderly patients from Romania, using multiple assessment tools to evaluate their effectiveness and agreement. Additionally, the study explores the relationship between frailty and socioeconomic factors, providing insights into the local determinants of frailty. To contribute to these efforts, we propose the GerEVal Napoca Frailty Index (GVN-FI), a new tool designed to meet the specific needs of the Romanian healthcare context and facilitate the better integration of frailty assessment into clinical practice.
Comments 4: Developing their own tool is an asset of the paper; however, it is composed of existing tools already available for clinical use. Furthermore, it is lengthy and relies on both subjective data and several objective measures. Nonetheless, the tool is more time-consuming to implement in clinical practice than other available frailty assessment tools in the literature.
Response 4: We appreciate your recognition of the GerEVal Napoca Frailty Index as an asset of the study and the value it adds to clinical practice. Below, we address your concerns regarding its composition and implementation:
1.Use of existing tools: While the GerEVal Napoca Frailty Index incorporates elements from existing frailty assessment methodologies, its design is based on a deficit accumulation framework tailored to the specific context of the Romanian healthcare system. By integrating both subjective and objective measures, we aimed to create a comprehensive tool that bridges gaps observed in the application of other frailty indices in clinical settings.
2.Time-consuming implementation: We acknowledge that the GerEVal Napoca Frailty Index may require more time to administer compared to some existing tools. However, this trade-off is deliberate. The index prioritizes a holistic evaluation that captures the multifaceted nature of frailty. Importantly, we found that much of the required information can be gathered during routine clinical assessments, minimizing additional workload for clinicians. To address implementation barriers, we are currently exploring ways to streamline the process, such as integrating the index into electronic health records or creating digital tools to support data collection and analysis.
3.Comparison with existing tools: We have highlighted in the manuscript that while shorter tools are available, they often focus on specific dimensions of frailty and may not fully account for the cumulative impact of deficit accumulation. The GerEVal Napoca Frailty Index, in contrast, allows for a broader, more personalized evaluation that aligns with the principles of person-centred care and facilitates tailored long-term management plans.
4.Adaptability to clinical practice: To enhance the utility of the GerEVal Napoca Frailty Index in clinical settings, we recommend its adoption in contexts where detailed geriatric evaluations are already standard practice, such as during comprehensive geriatric assessments. We believe that the added value it provides—particularly in identifying high-risk patients and optimizing resource allocation—justifies the time investment required for its application.